# The Interaction between Morbidity and Nutritional Status among Children under Five Years Old in Cambodia: A Longitudinal Study

**DOI:** 10.3390/nu11071527

**Published:** 2019-07-05

**Authors:** Gabriela Hondru, Frank T. Wieringa, Etienne Poirot, Jacques Berger, Somphos V. Som, Chan Theary, Arnaud Laillou

**Affiliations:** 1Reproductive and Child Health Alliance, No. 160 Street 71, Tonle Bassac, Chamkar Mon, P.O.Box 2471, Phnom Penh 12100, Cambodia; 2UMR-204 Nutripass, Institut de Recherche pour le Développement, IRD/UM/SupAgro, 34390 Montpellier, France; 3United Nations Children’s Fund (UNICEF), Integrated Early Childhood Development, Exchange Square, 5th Floor, No. 19&20, Street 106, Sangkat Wat Phnom, Khan Daun Penh, Phnom Penh 12100, Cambodia

**Keywords:** stunting, wasting, anthropometric failure, morbidity, children under five, acute illness

## Abstract

Even though limited evidence is available, the relationship between morbidity and under-nutrition among children under-five is likely to be a strong two-way association. This study aims to explore this vicious cycle by employing longitudinal data of four periods within a 24 month follow-up, whereby morbidity was captured between two subsequent anthropometric measures. Malnutrition was classified according to z-scores of anthropometric measures and morbidity by number of sick days experienced in between. Mixed-effects models were used to assess this relation, where dependency of morbidity and nutritional status were interchanged; models were adjusted for province, age, gender, wealth index score, maternal education level, diet, and Water, Sanitation, and Hygiene indicators. Stunting and wasting prevalence were 29.9% and 8.9%, respectively, where 21.3% of the children had multiple anthropometric failures. Children identified as wasted were 35% more likely to experience prolonged illness periods (OR: 1.35, 95% CI: 1.02–1.56). Those experiencing high proportion of sick days were found to be 64% more likely to become stunted (OR: 1.64, 95% CI: 1.18–2.29). This study suggests that the link between wasting and stunting could be partly explained by acute illness, where wasting increases the likelihood of prolonged episodes of illness, which increases the risk of stunting.

## 1. Introduction

Despite progress made over the past several years, malnutrition still represents a global health priority among children under five years of age. Stunting, or being too short for one’s age, is indicative of chronic under-nutrition, and is defined as two standard deviations below average height. The 2018 Global Nutrition Report estimated that 22.3% of all children under five years of age are affected by stunting, while the acute form, usually referred to as acute malnutrition or wasting, affected 7.5% of under-five children. Both stunting and wasting are associated with increased risk of mortality, with 10.5 million deaths in this age range attributed to under-nutrition [1]. Under-nutrition is not only associated with higher risk of morbidity and mortality but also has other long-term consequences, such as suboptimal cognitive development, lower economic productivity, and lower intellectual ability [2,3,4]. In Cambodia, stunting is estimated to affect 32% of children under five, while wasting and underweight affect 10% and 24% of children in this age group, respectively. These numbers highlight the high burden of malnutrition in the country [5].

Besides the presence of edema, acute malnutrition is identified by anthropometric failure of the weight-for-height Z-score (WHZ) at a level below −2 standard deviations (SDs) and/or a mid-upper arm circumference (MUAC) below 125 mm [6]. Both stunting and acute malnutrition are associated with an increased risk of infectious diseases; with episodes of symptoms such as diarrhea, fever, and breathing difficulties; and with longer periods of recovery [7,8,9]. So far, researchers have had difficulties establishing which form of under-nutrition has the most impact on the risk of acute illness; yet, evidence shows that concurrent types of malnutrition increase the risk of morbidity [10,11]. Fentahun et al. found that children with multiple forms of under-nutrition, defined as two or more anthropometric failures, were 2.6 times more likely to experience a period of acute illness [12]. Despite the increasing amount of evidence linking concurrent forms of malnutrition to morbidity and mortality, coexisting anthropometric deficits are, in general, overlooked and not reported, making evaluation of the whole under-nutrition panorama difficult [12,13].

For decades it has been known that morbidity is associated with faltered linear growth [12,13,14] as immune system responses and recovery are prioritized over linear growth during illness [15,16,17]. In addition, the loss of appetite and supplementary nutrient requirement experienced during illness contribute to a higher risk of becoming malnourished [8,18,19]. In a study conducted in India, the presence of illness in the past month increased the risk of developing any form of under-nutrition by 3.5 times [20], while in a study sample of Ethiopian children, an episode of diarrheal diseases in the past two weeks doubled their likelihood of becoming wasted [14]. Although catch-up growth is often seen after recovery, repeated episodes of illness and their increased duration are likely to have long-term effects on child linear growth.

Hence, a vicious cycle between morbidity and poor nutritional status exists where children with a poor nutritional status have an increased risk of experiencing infections with increased duration, consequently negatively influencing child growth. This study aimed at finding further evidence of this vicious cycle by fully exploring the two-way relationships between morbidity and malnutrition while considering multiple anthropometric failures. First, we assessed the relationship between the nutritional status of young Cambodian children and the occurrence and duration of acute illness episodes experienced after the taking of anthropometric measures. Secondly, we analyzed links between morbidity assessed through the time spent being ill and subsequent changes in anthropometric indicators.

## 2. Materials and Methods

### 2.1. Study Design and Sample

For this study, we analyzed longitudinal data from the Cambodian “MyHealth” program, initiated in 2016 by the Cambodian Ministry of Health, UNICEF, and the French Institute of Research for Development. During this continuing program, data are gathered on the health, use of health services, and nutritional status of young children and their mothers to evaluate the impact of national health and nutrition programs. The study area covers six districts in three provinces: Phnom Penh (one district), Kratie (two districts), and Ratanakiri (three districts). The latter two provinces are positioned in the north-east of Cambodia with a high proportion of rural inhabitants with low socio-economic status, while the district in Phnom Penh houses a poor semi-urban population with improved access to higher education, sanitation, and health services.

The study, shortly named MyHealth, received ethical approval under the name of “The Cambodian Health and Nutrition Monitoring Study” with file number 117/NECHR from the Cambodian National Ethical Committee for Health Research, National Institute of Public Health, Ministry of Health, Cambodia. 

Village health support groups and local midwives facilitated the recruitment process by listing all pregnant women and mothers/caregivers of children under the age of two years in the study area districts. Eligible mothers and infants were invited to participate in the MyHealth study in order on the list until the minimum sample size was reached. During the study, additional children were recruited. Also, children delivered from pregnant women recruited in MyHealth were recruited into the study. Informed consent was obtained from all participants. Consent for child participation was obtained through the adult primary caregiver (mostly the mother). All candidates were informed about the voluntary aspect of participation and the possibility to withdraw the consent at any given point. Children remained in the study until the age of five years. The minimum sample size was set to 1200 children per province, expecting a reduction in prevalence of stunting during a 24 month period of follow-up of 6% (from 32% to 26%) with a precision of 3% and an expected dropout rate of 20%.

The current study uses data from the baseline and follow-up Rounds 1 to 5. Data were collected using a tablet-based questionnaire covering information on anthropometric measures; socio-economic status; diet; and Water, Sanitation, and Hygiene (WASH) indicators. In September 2016, starting just after follow-up Round 1, regular morbidity monitoring visits were initiated with morbidity recalls each month in the first year and approximately every two months during the second year. In Phnom Penh, morbidity monitoring visits started at follow-up Round 2 (Figure 1). Changes in the program structure and adaptation to different field site conditions caused some irregularities in data collection during the fourth period. Data for analysis are from participants who were present during at least two succeeding follow-up rounds during which anthropometric measures and morbidity indicators were recorded and gender and date of birth were collected correctly. The eligible sample size for each period is illustrated in Figure 1.

### 2.2. Variables

Anthropometric measures including weight and height were collected in duplicate to the closest multiple of 100 gm and 1 mm, respectively. The tools for measurement were a SECA UK digital scale and a horizontal length scale with a moveable head board supplied by UNICEF Copenhagen. The average weight and height values were used to calculate z-scores of height-for-age (HAZ), weight-for-age (WAZ), and weight-for-height (WHZ) based on WHO Growth Child standards 2006 [6]. Outliers (WHZ and WAZ below −5 or above +5 and HAZ below −6 or above +6) were converted to missing values. The complexity of malnutrition was defined by concurrent cases of stunting (HAZ < −2 SDs), wasting (WHZ < −2 SDs), and/or underweight (WAZ < −2SDs) being categorized in either two or three anthropometric failures [12,13].

Morbidity monitoring visits collected information on acute illness experienced in the past 14 days. Severity was recorded as “No illness”, “Mild”, “Moderate”, or “Severe”, as well as whether the child had recovered or was still ill. If a child had been ill, questions were asked about the symptoms (“Diarrhea”, “Coughing”, “Fever”, “Difficulties breathing”, and “Other”) and the number of days the child had been ill. The morbidity experienced between two subsequent follow-up rounds was measured through the proportion of sick days, an indicator for the burden of all-cause morbiditycalculated from the declared number of days of the child being ill divided by the length of the recall period of 2 weeks. The mean value was then used and a criterion of having attended at least one morbidity monitoring visit during each period. For interpretation purposes the indicator was treated as a scale variable. The first and last quintiles were classified respectively as “Low” and “High” proportions of number of days ill, while the remaining quintiles were coded as “Medium”. The reference group was the “Low” category, which also included all the children who experienced no days of illness.

Information about province, child age, and gender were considered as basic characteristics of the participants. Several other co-variables were taken into account in the statistical methods used, including maternal educational level (categorized in three levels: ”No education or informal”, “Primary education”, and “Secondary and higher education”); household wealth index score (calculated according to Filmer and Pritchett through principal component analysis of the household assets and employment) [21]; hygienic environmental conditions of the household (measured through a composite score of water, sanitation, and hygiene -WASH- practices according to the Cambodian socio-economic survey, referred to as the National Child Sensitive composite score-CSES-) [22]. The later indicator covered source and treatment of household water, type of toilet and strategy of disposal of child feces, and habits of handwashing after handling child feces; each one of aforementioned was equality weighted based on a maximum value of 1 and points were offered for improved conditions. As WASH information was collected three times, household’s average score was used. Through this method, imputation was avoided.

On a similar note, diet was considered using “Appropriate Daily Feeding practices” (ADF) as described by the National Nutrition Guidelines [23]. Twenty-four-hour recall information was used for the construction of related indicators, namely, type of breastfeeding, frequency of feedings (breast milk and complementary foods), and quantity of food consumed [23]. Feeding practices were considered for the following age groups: (I) 0–5.99 months, (II) 6–8.99 months, (III) 9–11.99 months, (IV) 12–23.99 months, and (IV) 24 months+ [24]. Appropriateness of feeding practices was measured at each follow-up round through a binary value of appropriate (0) or not appropriate (1) and further used in the analysis as: Appropriate at all times (0), Sometimes appropriate (1), or Not appropriate (2).

### 2.3. Statistical Methods

The sample population of the study was first described for each follow-up, indicating changes over time in the prevalence of under-nutrition and z-score means. Thereafter, exploratory analysis of co-variables was conducted to show the distribution between those with and those without anthropometric failure. The proportionality between groups was tested with Pearson’s chi-square. The tests for continuous variables were separated into parametric and non-parametric methods after the normality of distribution was tested with the Lilliefors test. For parametric methods, ANOVA (one-way analysis of variances) with Tukey’s post hoc group testing was used for multiple comparisons, and for non-parametric methods, Kruskal–Wallis testing with a post hoc Wilcoxon test with Holm correction was used. Comparison of the mean of two samples used either the Wilcoxon rank-sum test for non-parametrical variables or the *t*-test for parametrical variables. Pearson’s correlations were performed between anthropometric measures and proportion of days ill to get an indication of a potential association.

The repeated measures were considered by a two-level random effect matching (1) individual-specific, child unique identification number measures in accordance with (2) the consecutiveness of data specific periods. Mixed-effects logistic regression models were used to calculate adjusted odds ratios of the impact of poor nutritional status on the magnitude of the proportion of days ill experienced in the period after the anthropometric assessment. Linear mixed-effects models were further used to estimate changes in the proportion of sick days, as a variable on a continuous scale, by nutritional status assessed at the beginning of the period. These methods are considered appropriate to analyze repeated measures by being subject-specific, allowing for within- and between-subject differences, and the random effects permitting for differences in timing. The vicious cycle was further explored via changes in nutritional status and z-scores measured at the end of a period and was considered an outcome influenced by independent variables. Linear and logistic mixed-effect models were used to estimate subsequent changes in anthropometric indices and the risk of becoming undernourished associated with a 1% increase in proportion of sick days and levels of magnitude of this variable.

The adjusted odds ratios and estimates were reflected by the fitted value with restrictive maximum likelihood with a 95% confidence interval (CI). All presented models were adjusted for age (1 month increase), sex (reference group: male), province (reference group: Kratie), maternal educational level (reference group: no education or informal), wealth index score (0.5 unit increase), CSES composite index (0.1 unit increase), and ADF index (reference group: appropriate daily feeding practices at all times). The covariation matrix was unstructured. The choice of best models and adjustments, including testing of interactions, was used based on the Akaike Information Criterion (AIC) and Bayesian Information Criterion (BIC).

The statistical analysis was performed in R version 3.4.0 by employing relevant packages [25,26,27]. Significance was considered at the *p*-value of 0.05. No imputation method was used.

## 3. Results

During the follow-up rounds, new children were included (352 at follow-up Round 2, 182 at follow-up Round 3, and 317 at follow-up Round 4), resulting in a sample of 5641 children: 1860 from Phnom Penh, 2072 from Kratie, and 1702 from Ratanakiri. The average drop-out rate was 6.5% (4.8%–8.2%), equivalent to an average of 220 to 390 cases per follow-up covering all three sites. Anthropometric measures were available from 3618 children at follow-up Round 1; 3448 children at follow-up Round 2; 3355 at follow-up Round 3; 3980 at follow-up Round 4; and 4140 at follow-up Round 5. Considered together with the morbidity information collected between two subsequent follow-ups, the sample population available for further investigations was 2683 children for Period 1; 3200 children for Period 2; 3957 for Period 3; and 3519 for Period 4. For Phnom Penh only, no morbidity information was collected between follow-up Rounds 1 and 2; hence this period is not included in the present study.

The study population was composed of 50.6% males and 49.4% females, starting at a mean age of 13.7 (+/− 7.8) months. Overall, the prevalence of under-nutrition was 29.9% for stunting, 8.9% for wasting, and 24.1% for underweight, with a significant proportion of children having at least two anthropometric failures (21.3%). Table 1 shows the differences in the prevalence of malnutrition and z-score distribution between the first and last follow-up rounds for which morbidity information is available. The proportion of children with no anthropometric failure decreased from 66.4% to 56.4%, with a 5.9% increase in the prevalence of multiple anthropometric failures (two or three), from a prevalence of 20.1% to 26%. The prevalence of stunting increased from 25.7% to 35.5%; the prevalence of underweight increased from 22.8% to 29.2%; while the proportion of children affected by wasting remained at 9.4% between the beginning and last follow-up rounds. The calculated proportion of days ill decreased between the first and last periods. For Period 1, an average of 22.2% of days of the child being ill was recorded (+/− 17.7), while for Periods 2 to 4, the respective averages were 21.7% (+/−16.7), 20.8% (+/−18.3), and 17.4% (+/−15.4). The calculated correlations between the z-scores of anthropometric measures, including WHZ and the proportion of sick days, show a weak negative relation (−0.05) at a *p*-value below 0.05.

Children with no anthropometric failures and those with at least one anthropometric deficit according to the WHO 2006 growth reference cut-toff points are compared in Table 2. One or more anthropometric failures were observed in 37.1% of all observations, with the prevalence being higher among males. Furthermore, households of children with at least one anthropometric failure had lower wealth index, lower WASH score, poorer feeding practices, and lived in the northeastern region of Cambodia (Ratanakiri followed by Kratie). In terms of the indicator for morbidity, those without any anthropometric failures had slightly fewer sick days (20.1% +/− 17.1) than those with at least one anthropometric failure (20.98% +/− 17.3), while the proportion in magnitude reveals that those with no anthropometric failures had shorter episodes of illness.

Logistic and linear mixed-effects models were used to assess the potential impact of nutritional status on the incidence of acute illness and their duration (Table 3). The incidence of acute illness, considered as general morbidity, showed no significant results (results not presented). The status of being wasted increased the chances by 25% to experience a High proportion of sick days (adjusted odds ratio (AOR): 1.25, 95% CI: 1.02–1.56, *p*-value = 0.03). The odds seemed to be slightly amplified by the presence of all three anthropometric failures (AOR: 1.34, 95% CI: 1.00–1.82, *p*-value = 0.04). Stunting, on the other hand, seemed to have no effect on the number of sick days a child will experience in the next period.

Considered on a continuous scale, the impact of wasting is estimated to increase the proportion of sick days experienced during each period by 1.38% (95% CI: 0.47–2.28, *p*-value < 0.01). This is amplified by the presence of all three anthropometric failures to 2.24% (95% CI: 1.02–3.47, *p*-value < 0.01). When calculated based on a two-week recall period, an increase of 1.38% is represented by 0.20 extra days of being ill, and 2.24% is represented by 0.34 additional sick days.

Individual changes in z-score values were investigated in relation to the history of morbidity experienced in the period prior to the anthropometric assessment with the aim to estimate the effect of prolonged periods of sickness on the child’s growth and nutritional status (Table 4). The estimated change in HAZ for a 1% increase in the proportion of sick days was −0.002 (95% CI: [−0.004, 0.000]), while the estimate in WAZ was lower and borderline significant and the estimate in WHZ was negligible (WAZ change: −0.0002, WHZ change: −0.001). The magnitude of sick days was used to represent the importance of results. Children registering a High proportion of sick days were estimated to encounter a drop of −0.10 SDs in HAZ (95% CI: [−0.15, −0.05]), −0.06 SDs in WHZ (95% CI: [−0.1, −0.02]), and −0.03 SDs in WAZ (95% CI: [−0.08, 0.18]) as compared with the z-score changes experienced by children with a Low proportion of sick days. In order to calculate what this would imply in terms of the risk to become stunted in our study sample, adjusted odds ratios were calculated. Being categorized as experiencing a High proportion of sick days as compared to Low was associated with a 64% increase in the possibility of becoming stunted (AOR: 1.64, 95% CI: [1.18, 2.29], *p*-value < 0.01), 19% greater likelihood to become wasted (AOR: 1.19, 95% CI: [0.72, 1.97], *p*-value = 0.4), and 26% greater likelihood to develop more than one anthropometric failure (AOR: 1.29, 95% CI: [0.93, 1.72], *p*-value = 0.06).

## 4. Discussion

Acute illness experienced in early childhood, in terms of incidence, duration, and seriousness of the episodes, can be influenced by the child’s nutritional status. On the other hand, linear growth faltering and the risk of developing at least one anthropometric failure are impacted by periods of illness. Understanding the cycle between acute illnesses and malnutrition is especially important for young children, as the consequences could affect both the short-term survival and the long-term well-being and development of the child. This study aimed at characterizing the two-way relation between under-nutrition, defined through forms and complexity, and morbidity, described by the duration of acute illness episodes. Even though it is relevant for countries with high prevalence of under-nutrition to assess the complexity of under-nutrition, this is the first study to our knowledge to identify and report multiple anthropometric deficits among young children in Cambodia.

Our results show that anthropometric failure of WHZ puts the child at a significantly higher risk of experiencing a prolonged period of acute illness. Even though wasting seems to be the main form of malnutrition associated with all-cause morbidity, all three anthropometric measures were found to increase these odds, proving once more that children with multiple anthropometric deficits should be considered as the most vulnerable group for acute illnesses.

Other studies not as specific in terms of the types of malnutrition have concluded that under-nutrition in general increases the risk and severity of infections [8,15,18,28]. This study strengthens the evidence that wasting, alone or in the presence of other anthropometric failures, impacts the risk of having prolonged periods of acute illness. The current explanation for this relation may be that a state of malnutrition, especially a low weight-for-height, weakens immune functions not only by affecting the integrity of the skin and mucosal barrier but also by impairing both humoral and cell-mediated immunity [19,29,30]. Epidemiological evidence suggests that increased risk of infections is correlated with severe anthropometric failure (−3 SDs) of both HAZ and WHZ, while severe anthropometric failure increases the risk of diarrhea frequency by 37% and the average duration of diarrheal episodes by 73% [15,31]. The findings of the present study show that even moderate levels of WHZ failure increase the duration of acute illness episodes, while multiple anthropometric measures add to this effect. More than half of the children (57%) registered at least one day of being ill at each morbidity monitoring visit, while a summary between two follow-ups revealed that 82.6% of the participants reported at least one episode of acute illness during each follow-up period. Our study found that a high proportion of sick days influences the risk of becoming stunted at a significant adjusted odds ratio of 1.64 (95% CI: 1.18–2.29, *p*-value < 0.05). This is further associated with an estimated loss of −0.10 SDs in HAZ (95% CI: [−0.15, −0.05], *p*-value < 0.05). The effect of prolonged episodes of acute illness on weight-for-height and weight-for-age is also visible, but to a smaller degree (WHZ: −0.06 SDs and WAZ: −0.03 SDs). A majority of existing evidence links episodes of diarrhea with poor nutritional status, while other symptoms or their duration are usually overlooked. Checkley et al. highlighted in a multi-country analysis that the risk of stunting increases by 13% at each fifth episode of diarrhea, while 25% of the incidence of stunting can be attributed to five or more episodes of diarrhea before the age of 24 months [32]. One study that pooled data from seven sites reported similar findings to our study, where a gap of 0.10 SDs in HAZ was identified between those with moderate-to-high occurrence of diarrhea and the control group [29]. Although Richard et al. showed that weight loss associated with diarrhea was only temporary, the restrictive effect on height may be noticeable only at a long-term follow-up [33]. Even though our study does not separate by type of disease, during an acute illness the response of the immune system is prioritized above child linear growth and weight gain. Furthermore, acute illness impacts the appetite of the child and thus restricts energy and nutrient intake as well as their bioavailability [8,18,19]. Hence, acute illness is a long-term determinant of population height and chronic malnutrition.

The prioritization of the treatment of severe acute malnutrition by public health care providers is explained by the substantial higher risk of case fatality as compared to chronic malnutrition. The latest evidence highlights the interaction between forms of under-nutrition and the difficulties of children with at least one anthropometric measure to obtain a long-term optimal nutritional status, pointing towards the benefits of an integrated approach to fight malnutrition [10,11,12,34]. Providing timely treatment, starting with moderate levels of under-nutrition, would have the potential to minimize the consequences of malnutrition and all related costs. This study brings further evidence and arguments to this debate. The findings of the present study show that even at moderate levels, acute malnutrition (−2 SDs of WHZ) impacts the duration of acute illness episodes, which then affect linear growth and increase the risk for stunting. Indeed, our study also suggests that the duration of acute illness might be one of the factors explaining the interaction between acute and chronic malnutrition. As it is outside of the aim of this paper, we encourage future research on the interaction between types of malnutrition with a potential mediator effect from morbidity measured through the length of acute illness episodes.

Besides the importance of the findings and strengths linked to the methodology, limitations should be also acknowledged. It is debatable whether the burden of morbidity can be assessed only through recall of the length of acute illness episodes. Even though the type of symptoms and chronicity could offer further understanding, this study could not add this perspective for multiple reasons. It was observed that for a vast majority of children, multiple symptoms were reported. Furthermore, the completeness of the data potentially limited the perspective on chronic cases of illness and suggested other serious consequences, such as hospitalization. The recall period of two weeks might raise difficulties for caregivers to accurately estimate the number of days ill. Even though data collectors were trained, measurement biases might arise from a limited knowledge level. Therefore, we acknowledge the limitations that our study presents, especially in terms of assessing morbidity, and we would like to encourage future research to help find an easily implemented tool to assess morbidity in communities.

## 5. Conclusions

This study is one of the first to evaluate the two-way relation between morbidity and under-nutrition by using data collected during a longitudinal study. The prevalence of under-nutrition was found to be high when compared with other countries and global rates, while a significant proportion of children was identified with multiple anthropometric failures. Children under five years of age with acute malnutrition were estimated to have an increased risk to experience a prolonged period of acute illness, most likely because of diminution of the immune responses, requiring more time to tackle the cause of the acute illness. A high proportion of days ill, in its turn, was associated with higher risk of stunting and significant decreases in z-scores for height-for-age and weight-forheight. In a setting with a high prevalence of stunting and of concurrent forms of under-nutrition, timely treatment of moderate acute malnutrition has not only the potential to reduce the burden of acute illness but also the potential to reduce the prevalence of stunting. Further research would represent an opportunity to deepen understanding of the potential role of morbidity in the interaction between stunting and wasting.

## Figures and Tables

**Figure 1 nutrients-11-01527-f001:**
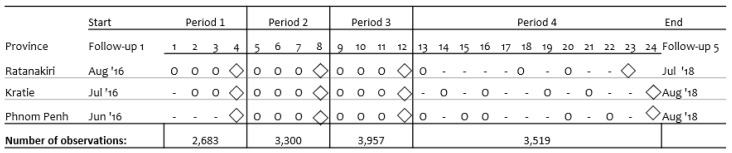
Timeline of the study represented by the period between follow-ups and including the sample size available for each period. ◊ Follow-up visit; o Morbidity monitoring visit; Number of observations: Observations that have both anthropometric and morbidity information.

**Table 1 nutrients-11-01527-t001:** Prevalence of malnutrition and population change in z-scores between first (follow-up Round 1 for Kratie and Ratanakiri and follow-up Round 2 for Phnom Penh) and last follow-up (follow-up Round 5).

Study Period	Follow-up Round 1/2 *n* = 3580 % (*n*)	Follow-up Round 5 *n* = 3519 % (*n*)	*p*-Value
Mean age in months	13.7	31.7	
Females	49.7% (1786)	49.2% (1758)	
**Types of malnutrition**			
Stunting	25.7% (4004)	35.3% (1245)	<0.01
Wasting	9.4% (335)	9.4% (332)	0.8
Underweight	22.8% (818)	29.2% (1027)	<0.01
**Complexity of under-nutrition**			<0.01
None	66.4% (2,377)	56.7% (1996)	
One anthropometric failure	13.5% (483)	17.3% (608)	
Two anthropometric failures	15.9% (571)	21.3% (748)	
Three anthropometric failures	4.2% (149)	4.7% (166)	
**Z-scores**	mean (SD)	mean (SD)	
HAZ	−1.23 (1.20)	−1.55 (1.12)	<0.01
WHZ	−0.70 (1.03)	−0.79 (0.99)	0.05
WAZ	−1.18 (1.09)	−1.43 (1.04)	<0.01

HAZ, height-for-age; WHZ, weight-for-height; WAZ, weight-for-age. One anthropometric failure: child is either stunted, wasted, or underweight according to WHO 2006 growth reference cutoffs. Two anthropometric failures: child has two concurrent types of malnutrition. Three anthropometric failures: the child is stunted, wasted, and underweight.

**Table 2 nutrients-11-01527-t002:** Distribution of all observations by the absence or presence of anthropometric failure(s).

Variables	No Anthropometric Failure *n* = 8927 *n* (%)/mean (SD)	At Least One Anthropometric Failure *n* = 5268 *n* (%)/mean (SD)	*p*-Value
**Age of child(months)** Mean +/− SD	21.11 +/− 12.2	24.50 +/− 11.4	<0.01
**Sex**			<0.01
Female	4357(61.3%)	2748 (38.7%)	
Males	4570(64.4%)	2520 (35.5%)	
**Province**			<0.01
Ratanakiri	2661 (53.7%)	2294 (46.3%)	
Kratie	4019 (64.9%)	2171 (35.1%)	
Phnom Penh	2661 (73.7%)	803 (26.3%)	
**Maternal education level**			<0.01
None or informal	1482 (50.3%)	1466 (49.7%)	
Primary	2849 (61.9%)	1749 (38.1%)	
Secondary and more	2661 (70.0%)	1140 (30.0%)	
**Wealth index score**			<0.01
Mean +/− SD	0.04 +/− 1.5	-0.47 +/− 1.3	
**CSES composite score**			<0.01
Mean +/− SD	0.75 +/− 0.2	0.69 +/− 0.2	
**Appropriate feeding**			<0.01
Appropriate	2411 (66.4%)	1222 (33.6%)	
Sometimes appropriate	4337 (62.0%)	2656 (38.0%)	
Not appropriate	1370 (58.3%)	979 (42.7%)	
**Proportion of sick days**			<0.01
Mean +/− SD	20.01 +/− 17.1	20.98 +/− 17.3	
**Magnitude of proportion of sick days**			<0.01
Low	1670 (67.7%)	796 (32.3%)	
Medium	4984 (61.3%)	3141 (38.7%)	
High	1683 (60.9%)	1079 (39.1%)	

**Table 3 nutrients-11-01527-t003:** The impact of nutritional status on the proportion of sick days experienced in the next period by calculated adjusted odds ratios (AORs) and fitted estimates of mixed-effect models.

Variable:	Magnitude of Proportion of Sick Days		Proportion of Sick Days	
	AOR ^a^	95% CI	Estimate ^b^	95% CI
Stunting	1.06	(0.86, 1.27)	0.74	(−0.03, 1.51)
Wasting	1.25 *	(1.02, 1.56)	1.38 *	(0.47, 2.28)
Complexity of under-nutrition				
None	REF.	REF.	REF.	REF.
One anthropometric failure	0.95	(0.77, 1.19)	−0.02	(−0.94, 0.91)
Two anthropometric failures	1.05	(0.84, 1.30)	0.51	(−0.42, 1.45)
Three anthropometric failures	1.29	(0.96, 1.72)	2.24 *	(1.02, 3.47)

^a^ Model adjusted for fixed effects as follows: age, gender, wealth score index, CSES average composite score, Appropriate Daily Feeding practices (ADF) average score, and province. Two-level random effects: (1) the period; (2) the child ID. The outcome is the magnitude of the proportion of sick days. ^b^ Model adjusted for fixed effects as follows: age, gender, wealth score index, CSES average composite score, ADF average score, and province. Two-level random effects: (1) the period; (2) the child ID. The dependent variable was the proportion of sick days as a continuous variable. * *p*-value below 0.05.

**Table 4 nutrients-11-01527-t004:** The effect of proportion of sick days on nutritional status and z-score changes by calculated adjusted odds ratios (AORs) and fitted estimates (LMER) of mixed-effect models.

Variable:	HAZ ^a^ Estimates (95% CI)	WHZ ^a^ Estimates (95% CI)	WAZ ^a^ Estimates (95% CI)	Stunting ^b^ AOR (95% CI)	Wasting ^b^ AOR (95% CI)	Multiple AF ^b^ AOR (95% CI)
Magnitude of proportion of sick days						
Low	REF.	REF.	REF.	REF.	REF.	REF.
Medium	−0.04 (−0.08,0.01)	−0.05 * (−0.09,−0.02)	−0.01 (−0.04,0.03)	1.17 (0.89,1.53)	1.14 (0.76,1.73)	1.03 (0.79,1.32)
High	−0.10 * (−0.15,−0.05)	−0.06 * (−0.11,−0.02)	−0.03 (−0.08,0.18)	1.64 * (1.18,2.29)	1.19 (0.72,1.97)	1.26 (0.93,1.72)

^a^ Model adjusted for fixed effects as follows: age, gender, wealth score index, CSES average composite score, ADF average score, and province. Two-level random effects: (1) the period; (2) the child ID. The outcome was the z-score as a continuous variable. ^b^ Model adjusted for fixed effects as follows: age, gender, wealth score index, CSES average composite score, ADF average score, and province. Two-level random effects: (1) the period; (2) the child ID. The dependent variable was stunting, wasting, or two or more forms of anthropometric deficits. * *p*-value below 0.05.

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
