# Peer review of "The Interaction between Morbidity and Nutritional Status among Children under Five Years Old in Cambodia: A Longitudinal Study"

_nutrients, 2019, doi:10.3390/nu11071527_

Round 1
Reviewer 1 Report
While this is undoubtedly interesting and important data and the findings seem straightforward, its presentation is poor so that it is very difficult to identify exactly how the study was carried out.
Major issues
1. Study design is poorly described in terms of the overall length of the study or the timing of each phase of the study, i.e. the length of each period, and the time (in months) between follow ups. i.e. it states Lines 88-90 “Data collection was organized in Follow-up visits, from Baseline to Follow-up 5, and, after Follow-up 1, regular morbidity monitoring visits conducted each month in the first year and each second month afterwards”. And “Morbidity monitoring visits collected 14-days recall information”. This presumably means that Periods 1-3 each included 4 monthly morbidity measurements (i.e. 4 months each, 12 months total) with period 4 including 4 2-monthly morbidity measurements (i.e. 8 months total) giving 20 months for the overall study. Is this correct?
It is also not clear what the timing of the 5 follow up periods shown in table 1 were. Was follow up 1(FU1) the same as baseline since according to table 2 the data shown was collected at the end of each period (Period 1-Follow-up 2, Period 2-Follow-up 3, Period 3- Follow-up 4, Period 4-Follow-up 5)?. However it is not clear why FU2, FU3 and FU4 overlapped the four periods.
2. If FU1 is baseline it would be valuable to include a table of the baseline data. This is particularly important since according to table 2 there were no changes in the extent of stunting, wasting or underweight over the study. Given that this was a longitudinal study it is quite surprising that the authors do not comment on this.
Presentation of results.
Table 2: Proportion of sick days: what do the values in brackets refer to?
Table 3 It is not clear what the p values refer to in relation to sex (other than more females and males have no anthropometric failure compared with at least one).
For Appropriate feeding is 0.47 +/- 0.4 worse than 0.42 +/- 0.4?
For Proportion of sick days are the two values (20.01 +/-17.1 and 20.98+/-17.3) significantly different at the<0.01 level?
Author Response
---

Reviewer 2 Report
Thank you very much for your contributions to the field. "The interaction between morbidity and the nutritional status among children under-five in Cambodia: A longitudinal study" is an important study that attempts to novelly link the bidirectional nature of morbidity and malnutrition in children under 5 in Cambodia. The impact of this study could be significant, but there are major methodological limitations that question whether this study can achieve it's objectives. The first major concern is that participants were observed en-mass rather than based on an adequate timeframe and number of observations. Additionally, it is not clear whether this study is even able to assess changes in stunting/wasting given its methodological limitations. Please see my comments for further questions.

Author Response
---

Round 2
Reviewer 2 Report
The clarity of the study design and results for this work has been greatly improved with editing, and I am satisfied with justifications for the research presented. While there continue to be some methodological questions around quality of morbidity data, this paper presents important information despite its limitations. There are several grammatical changes between present and past tense in the paper that can still be improved throughout.
Author Response
Dear reviewer,
Thank you for your comments! It helped tremendously to improve the quality of the manuscript.
Best regards,
Gabriela Hondru